# Applications of minimally invasive multimodal telemetry for continuous monitoring of brain function and intracranial pressure in macaques with acute viral encephalitis

Henry Ma[1], Jeneveve D. Lundy[1], Emily L. Cottle[1], Katherine J. O'Malley[1], Anita M. Trichel[1], William B. Klimstra[1], Amy L. Hartman[1], Douglas S. Reed[1,2‡]*, Tobias Teichert[3,4‡]*

1 Center for Vaccine Research, University of Pittsburgh, Pittsburgh, Pennsylvania, United States of America, 2 Department of Immunology, University of Pittsburgh School of Medicine, Pittsburgh, Pennsylvania, United States of America, 3 Department of Psychiatry, University of Pittsburgh School of Medicine, Pittsburgh, Pennsylvania, United States of America, 4 Department of Bioengineering, University of Pittsburgh, Pittsburgh, Pennsylvania, United States of America

‡ DSR and TT are Joint senior authors on this work.
* dsreed@pitt.edu (DSR); teichert@pitt.edu (TT)

**Data Availability Statement:** The GraphPad files used to create the figures are available on figshare:

## Abstract

Alphaviruses such as Venezuelan equine encephalitis virus (VEEV) and Eastern equine encephalitis virus (EEEV) are arboviruses that can cause severe zoonotic disease in humans. Both VEEV and EEEV are highly infectious when aerosolized and can be used as biological weapons. Vaccines and therapeutics are urgently needed, but efficacy determination requires animal models. The cynomolgus macaque (*Macaca fascicularis*) provides a relevant model of human disease, but questions remain whether vaccines or therapeutics can mitigate CNS infection or disease in this model. The documentation of alphavirus encephalitis in animals relies on traditional physiological biomarkers and behavioral/neurological observations by veterinary staff; quantitative measurements such as electroencephalography (EEG) and intracranial pressure (ICP) can recapitulate underlying encephalitic processes. We detail a telemetry implantation method suitable for continuous monitoring of both EEG and ICP in awake macaques, as well as methods for collection and analysis of such data. We sought to evaluate whether changes in EEG/ICP suggestive of CNS penetration by virus would be seen after aerosol exposure of naïve macaques to VEEV IC INH9813 or EEEV V105 strains compared to mock-infection in a cohort of twelve adult cynomolgus macaques. Data collection ran continuously from at least four days preceding aerosol exposure and up to 50 days thereafter. EEG signals were processed into frequency spectrum bands (delta: [0.4 – 4Hz]; theta: [4 – 8Hz]; alpha: [8-12Hz]; beta: [12–30] Hz) and assessed for viral encephalitis-associated changes against robust background circadian variation while ICP data was assessed for signal fidelity, circadian variability, and for meaningful differences during encephalitis. Results indicated differences in delta, alpha, and beta band magnitude in infected macaques, disrupted circadian rhythm, and proportional increases in ICP in response to alphavirus infection. This novel enhancement of the cynomolgus

https://figshare.com/articles/Data_Set_for_PLoS_ONE_paper/12475877.

**Funding:** This research was funded by Defense Threat Reduction Agency (DTRA, https://www.dtra.mil/), grant number W911QY-15-1-0019 and The APC was funded by the same. Disclaimer: This project was sponsored by the Department of the Army, U.S. Army Contracting Command, Aberdeen Proving Ground, Natick Contracting Division, Ft. Detrick, MD under grant no. W911QY-15-1-0019, awarded to ALH, WK, and DSR. Any opinions, findings, and conclusions or recommendations expressed in this material are those of the author(s) and do not necessarily reflect the position or the policy of the government and no official endorsement should be inferred. Research was conducted in compliance with the Animal Welfare Act and other Federal statutes and regulations relating to animals and experiments involving animals and adheres to principles stated in the Guide for the Care and Use of Laboratory Animals, National Research Council, 1996. The facility where this research was conducted is fully accredited by the Association for Assessment and Accreditation of Laboratory Animal Care International. The funders had no role in study design, data collection and analysis, decision to publish, or preparation of the manuscript.

**Competing interests:** The authors have declared that no competing interests exist. The funders had no role in the design of the study; in the collection, analyses, or interpretation of data; in the writing of the manuscript, or in the decision to publish the results. None of the submitted work was carried out in the presence of any personal, professional, or financial relationships that could be potentially construed as a conflict of interest; none of the authors listed owns controlling interest in any of the vendors or suppliers listed in this manuscript.

macaque model offers utility for timely determination of onset, severity, and resolution of encephalitic disease and for the evaluation of vaccine and therapeutic candidates.

## Introduction

The genus Alphavirus of the family *Togaviridae* composes a group of linear, non-segmented positive-sense single-stranded RNA arboviruses found in many regions of the world. New World alphaviruses can invade the central nervous system (CNS) and cause severe encephalitis which can be diagnosed by fever, brain swelling, and neurological symptoms such as seizures and a diffuse slowing of brain activity across the physiologically relevant electroencephalography (EEG) frequency spectra [1, 2]. While Venezuelan equine encephalitis virus (VEEV) is only rarely lethal in humans, North American strains of Eastern equine encephalitis virus (EEEV) have a high mortality rate in confirmed cases (estimates range from 30–70% between outbreaks) and a high percentage of survivors have long-term neurological sequelae [3]. Laboratory-acquired infections have demonstrated a high infectivity of aerosolized VEEV and EEEV [4–6] with disease onset, clinical signs, and outcomes very similar to natural mosquito-borne infection. There is significant concern that VEEV or EEEV could be employed as biological weapons by rogue nations or terrorists, and these viruses are listed as select agents [7]. Because of these concerns, there is an urgent need for licensed vaccines and/or treatments.

Aerosol transmission of VEEV and EEEV does not comprise a natural route of transmission; because of disease severity and the lack of natural infections, the evaluation of safety and efficacy for potential medical countermeasures must proceed in animals in accordance with the FDA's "Animal Rule." The Animal Rule allows for the use of animals in pivotal efficacy studies to support licensure if the disease in the animal model sufficiently resembles human disease [8–10]. Cynomolgus macaques (*Macaca fascicularis*) in particular were among the first laboratory animals used for the study of equine encephalitis viruses, and have seen recent development efforts as a nonhuman primate (NHP) model for both VEEV and EEEV [3, 11–15]. In human cases of VEEV and EEEV, commonly reported clinical findings and symptoms include fever, elevated heart rate, malaise, anorexia, dehydration as well as classical symptoms of viral encephalitis: headache, neck stiffness, photophobia, and tremors. Insomnia is sometimes reported, and extremely severe cases can develop seizures and coma [16, 17].

We and others have previously shown that exposure of cynomolgus macaques to aerosolized VEEV or EEEV results in a disease that strongly resembles that reported during natural human disease caused by these viruses, including fever, ECG abnormalities such as QRS complex and QT-Interval abnormalities, and clinical signs that suggested neurological disease [12]. Exposure of macaques to aerosolized EEEV resulted in fatal disease at high doses, typically within 5–6 days. Elevated heart rates were also seen and a prominent neutrophilia that predicted outcome. At lower doses, no febrile response was seen and macaques survived exposure [3]. We recently reported ECG abnormalities seen with both VEEV and EEEV infection including a loss of heart-rate variability that coincided with the febrile response and might be useful in determining outcome, particularly for EEEV [14]. While these and other studies have provided important insights, they were also constrained by technical limitations that could impact inter-rater reliability and reproducibility. In particular, determining the onset and resolution of encephalitis often depends on subjective and intermittent veterinary staff observations, rather than continuous and objective measurements of brain-specific parameters that can capture and quantify the complex and multi-modal manifestation of encephalitic disease.

To address these limitations and to extract more data from costly NHP studies, we set out to develop a surgical method for the implantation of a radiotelemetry device for continuous monitoring of EEG and intracranial pressure (ICP) in awake, conscious macaques, and complementary methods for the collection and analysis of EEG and ICP data suitable for detecting significant changes indicative of CNS disease. This method constituted a (i) minimally invasive, (ii) multi-modal, and (iii) continuous monitoring of brain function during alphavirus infection. In this context, we define 'minimally invasive' as leaving the *dura mater* intact. This was considered important to prevent pathogens from entering the brain in a non-physiological way, as well as to mitigate the possibility of generating artifactual data. We define 'continuous' as no more than ten minutes' interruption of data collection per day. Uninterrupted recordings are important because pathological changes can manifest rather quickly at any point of the day or night. Furthermore, continuous recordings facilitate the extraction and modeling of circadian rhythms which is critical because (i) loss of circadian rhythms can present a meaningful sign of disease, and (ii) circadian fluctuations pose an important confounding variable against which to evaluate other disease-related changes of relevant physiological measures [3, 12–14]. The loss of circadian variation is likely caused by the effects of cytokines such as IL-6 on the biomolecular clock, and as such, circadian rhythms can serve as a surrogate marker of the inflammatory response. Although other neurological events detectable by EEG such as seizures are undoubtedly related to increases in cytokines or acute phase reactants, the biomolecular milieu that underlies a condition such as brain swelling also pertains to disruption of the circadian rhythm. This is particularly important for alphavirus research goals, especially with regard to the evaluation of pathological changes in response to infection on long-term timescales [18–21]. Finally, we define 'multi-modal' as including (i) a systemic measure of immune response and inflammation: temperature, (ii) an anatomical measure of brain swelling: ICP, and (iii) an electrophysiological measure of brain function: EEG. Literature from human clinical cases have demonstrated the appearance of abnormal EEG alongside radiography indicative of brain swelling [22]. Similarly, brain swelling and the putative rise in ICP are observed several days after the onset of alphavirus encephalitic disease in humans–continuous monitoring can thereby help to signal the need for timely intervention [23, 24]. More importantly, in the context of this animal model, continuous EEG/ICP monitoring can help to assess the safety and efficacy of medical countermeasures.

Based on these technological innovations, we wished to address several important scientific goals and research questions. (i) How stable are EEG and ICP responses over the course of many days? (ii) Does systematic circadian variation of EEG and ICP exist? (iii) How consistent are EEG and ICP responses between different macaques? With this in mind, the further pursuance of these questions in this animal model are essential to the process of determining whether EEG and ICP changes can serve as reliable biomarkers of the pathological state that might be encountered during an acute febrile encephalitis.

In this report, we detail methods for the implantation of telemetry devices to monitor EEG and ICP as well as for the collection and analysis of this data. Data from two mock-infected macaques highlight the stability of the measurements and suggest that they are well suited to detect pathological changes in EEG or ICP that are indicative of CNS disease. Furthermore, we present preliminary data from 10 macaques experimentally infected with alphaviruses that demonstrate the utility of these techniques in identifying pathologically relevant increases of ICP during acute viral encephalitis that correlate with the emergence of the systemic immune response and could predict the emergence of severe and highly specific episodes of altered EEG. This data provides a richer understanding of the disease course in macaques after aerosol exposure to VEEV or EEEV and the relevance to human disease and its resolution may be useful in the evaluation of efficacy of potential vaccines and therapeutics. A full characterization

of the macaque model of EEEV and VEEV will be covered in subsequent manuscripts currently under preparation; the purpose of this report is to evaluate whether deviations from EEG baseline frequency spectra occurred after EEEV/VEEV infection and the nature of these deviations.

## Materials and methods

### Telemetry materials and surgical supplies

Table 1 represents a parts list with details of manufacturers, part numbers, and specifications.

**Table 1. Materials list.**

| | Part | Manufacturer | Model/Catalog Number | Notes |
|---|---|---|---|---|
| **SURGERY** | Stereotaxic Frame Assembly Cat/Monkey | David Kopf Instruments | Model 1430-B | |
| | Upright Post and Clamp Assembly | David Kopf Instruments | Model 1725 | |
| | Intracellular Base Plate Assembly | David Kopf Instruments | Model 1711 | |
| | Copalite Varnish | Temrex Corporation | 200-400001V | 0.5oz Bottle |
| | Titanium Bone Screws | Crist Instruments Co., Inc. | 6-YXC-035 | |
| | Gold-Plated Amphenol Pin | DIGI-KEY | 609-2814-ND | 1mm Diameter |
| | Dental Acrylic | Lang Dental Mfg | 1230FIB | |
| | Drill | Veterinary Orthopedic Implants | 20020 | Hand Drill, Quick-Coupling Driver Handle |
| | Coupling Bits | Veterinary Orthopedic Implants | 10006 | 1.5mm to 2.0mm Quick-Coupling Bits |
| **TELEMETRY** | PhysioTel Digital RF Transmitter Model No. M01 | Data Sciences International | M01 | EEG & Temperature Only, ~40 day battery charge |
| | PhysioTel Digital RF Transmitter Model No. M11 | Data Sciences International | M11 | EEG/ICP, and Temperature ~40 day battery charge |
| | PhysioTel Digital RF Transmitter Model No. L11 | Data Sciences International | L11 | EEG/ICP, and Temperature ~105 day battery charge |
| | Ponemah Software Package | Data Sciences International | v.5.20 SP 8 | |
| | NeuroScore | Data Sciences International | v.3.0.0 | |
| | Communication Link Controller | Data Sciences International | 272-8000-001 | CLC Firmware version 0.1.28 |
| | Radiofrequency Telemetry Transciever | Data Sciences International | 272-9000-001 | Includes proprietary TRX-1 to Ethernet Cables |
| | Ambient Pressure Reference (APR-1) Module | Data Sciences International | 275-0020-001 | Measures ambient air pressure |
| | E2S-1 Module | Data Sciences International | 275-0248-001 | Communicates between Ethernet switch and APR-1 |
| | Telemetry Acquisition Computer | Lenovo | Thinkstation Desktop | 64-bit CPU |
| | Ethernet Switch/Router | Cisco | SF100D-16P | |
| | Cat6E Ethernet Cables | - | - | Color-coded if purchased from Data Sciences International |
| | Axis Closed Circuit IP Camera | Axis | 0436-001-01 / 0591-001-02 | Models M1144-L / M1155-L |
| | Noldus Media Recorder | Noldus | v.2.6 | |
| | MATLAB | Mathworks | R2018b | |
| | PRISM | GraphPad | v.8.0.0 | |

Part, manufacturer, and catalog numbers of materials used categorized into surgical and telemetry subsections. Notes section contains details on specifications as well as doses supplied, if applicable.

## Ethical statement

This work was approved by the University of Pittsburgh Institutional Animal Care and Use Committee (IACUC) under protocol numbers 16026773 and 17100664 and adhered fully to the Animal Welfare Act Regulations and the Guide for the Care and Use of Laboratory Animals [25]. Care and treatment of the macaques was in accordance with the guidelines set by the U.S. Department of Health and Human Services (National Institutes of Health) for the care and use of laboratory animals. All surgical procedures were performed under isoflurane gas anesthesia, and all efforts were made to minimize suffering. The University of Pittsburgh is fully accredited by the Association for Assessment and Accreditation of Laboratory Animal Care (AAALAC). The rationale for the development of a nonhuman primate (NHP) model of alphavirus encephalitis fulfills the necessity criteria outlined by the FDA Animal Rule [8, 10, 26].

## Study design and experimental animals

Studies were performed in 12 adult outbred cynomolgus macaques (*Macaca fascicularis*). Table 2 summarizes pertinent characteristics of the macaques used in this study. The experimental group consisted of six EEEV-infected macaques and four VEEV-infected macaques, while the control group consisted of two mock-infected macaques. Baseline telemetry data were collected for at least four days prior to exposure for all macaques. Macaques were randomly assigned to each group. These macaques were part of a larger cohort used for evaluating a number of biomarkers in use for marking clinical course and outcome of infection in macaques infected with VEEV or EEEV. We have previously reported on evaluation of ECG

**Table 2. Study cohort characteristics.**

| | Animal ID | Age (yr) | Sex (M/F) | Weight (kg) | Inhaled Dose $\log_{10}$ PFU | Neurological Disease (Y/N) | Time to Fever Onset (DPI) | Fever Duration (days) | Implant Type |
|---|---|---|---|---|---|---|---|---|---|
| MOCK | M115-16 | 6 | M | 4.6 | 0.0 | N | - | - | M11: EEG/ICP |
| | M116-16 | 5 | M | 5.2 | 0.0 | N | - | - | M11: EEG/ICP |
| VEEV | M111-18 | 6 | M | 7.2 | 7.5 | Y | 1.0 | 12.3 | L11: EEG/ICP |
| | M112-18 | 9 | M | 7.6 | 7.3 | Y | 1.0 | 8.3 | L11: EEG/ICP |
| | M115-18 | 5 | M | 6.7 | 7.2 | Y | 1.0 | 18.8 | L11: EEG/ICP |
| | M116-18 | 9 | M | 6.9 | 7.8 | Y | 1.0 | 12.1 | L11: EEG/ICP |
| EEEV | M120-16 | 5 | F | 3.8 | 7.0 | Y | 1.5 | 3.0 | M01 EEG |
| | M57-17 | 5 | M | 6.0 | 8.6 | Y | 2.3 | 2.8 | M01 EEG |
| | M58-17 | 5 | M | 5.5 | 9.0 | Y | 2.3 | 4.3 | M01 EEG |
| | M160-17 | 6 | M | 6.6 | 9.1 | Y | 3.5 | 3.6 | M11: EEG/ICP |
| | M1-19 | 2 | F | 2.3 | 8.0 | Y | 2.5 | 2.9 | M11: EEG/ICP |
| | M3-19 | 2 | M | 2.7 | 8.6 | Y | 2.5 | 2.6 | M11: EEG/ICP |
| MEAN | - | 5.4 | - | 5.4 | - | Y | - | - | - |

Age (yr), sex (M/F), weight (kg), $\log_{10}$ dose of virus (PFU), presence/absence of neurological disease (Y/N), time of onset (DPE) and total duration of fever (days).

data from this model [14]. For this report, with an expected incidence of 0% encephalitis or fever detectable by radiofrequency telemetry in mock-infected macaques and a >99% expected incidence of fever or encephalitis in macaques exposed to alphavirus, a 1:1 enrollment ratio, and Type I and II error rates of 0.05 and 0.80, respectively, we determined that our sample size required a minimum of two macaques per group. The enrollment of four and six macaques infected with VEEV and EEEV respectively allowed for an increased number of replicates for the purposes of validation. All macaques were euthanized either at the end of the study period (typically 28 days post-challenge) or if a humane study endpoint was triggered because the animals was moribund. Euthanasia was accomplished by initial sedation with 20mg/kg IM ketamine, followed by IV administration through the great saphenous vein of sodium nitroprusside mixed with 12mL of 0.9% normal saline and 200mg/kg of Beuthanasia phenytoin/pentobarbital for induction of respiratory arrest.

## Housing and husbandry

Cynomolgus macaques were singly housed in quad-occupancy metal cage banks with absorbent pellet bedding and food forage in a facility for these studies with excreta and food/fluid intake monitored daily. Environmental conditions prevailed at a constant 22˚C, with a weight-based diet formulation of Purina Monkey Chow and water *ad libitum*. The conditions under which the macaques were housed provided a binary day-night cycle, during which room lights were powered on for 12 hours beginning at 6:00AM each day, and powered off for 12 hours at 6:00PM. Enrichment consisted of food enrichment, puzzles and toys, as well as audio-visual stimulation in the form of several hours of television per day.

## Clinical observations

Subject welfare was assessed on a daily basis, at least once per day by a series of ordinal scales for: neurology, activity, and temperature [3, 12, 13]. All scoring scales ran from 1–6. The neurology scale accounted for signs of neurological disease such as involuntary tremor, gait imbalance, nystagmus, head pressing, seizures, and coma. Seizures were defined primarily by incidental human observation on welfare assessment and secondarily by visual review of recorded video data. The activity scale accounted for posture, facial expressions, responses to stimuli, and interactions with observers. Finally, the temperature scale accounted for core temperature set at standard deviations above or below mean core temperature to detect fever or hypothermia. The cumulative score determined by summing all three scales' scores was used to determine whether a macaque required more frequent observation or was at risk of imminent death which triggered the humane study endpoint. No analgesia or anti-seizure medications were employed to minimize potential impact on EEG and ICP data collection.

## Surgical procedure

On the day of surgery, each macaque was given atropine (0.04 mg/kg, intramuscularly (IM)) to reduce mucus secretions. Anesthesia was induced with ketamine hydrochloride (20 mg/kg, IM). In preparation for surgery, an intravenous (IV) catheter for 0.09% normal saline was inserted into the greater saphenous vein, and the macaques were intubated with a tracheal tube for oxygen (1.0–1.5 L/min) and gas anesthesia. The macaque's head, neck and upper back were shaved and the macaque was transferred to the surgical suite and maintained on gas anesthesia (Isoflurane 0.5–3.0%) throughout surgery. The macaque was positioned in a five-point stereotax to ensure positional stability during surgery. The incision sites were aseptically prepared with betadine and chlorhexidine scrubs, and draped with sterile drapes. Heart rate, respiratory rate, pulse oximetry, and core temperature were monitored continuously until the

end of surgery. Incisions were made to expose the surgical sites on the scalp and on the upper back between the scapulae slightly off midline.

A ~7cm long incision was made along the midline of the head from ~5mm posterior of the ocular ridge to the occipital ridge. Fascia and temporalis muscle were separated from the calvaria using sharp surgical spoons. Residual tissue was thoroughly scraped from the bone, and the skull was dried with sterile gauze. A 5cm long incision was then made between the scapulae on the macaque's upper back. Using blunt dissection, a subdermal pocket superficial to the trapezius muscle was formed of sufficient size to contain the telemetry transmitter (Data Sciences International (DSI) Model No. M01, M11, or L11). A tunneling rod was inserted in the caudal aspect of the cranial cut and tunneled to the pocket between the scapula. The two EEG lead wires and the ICP pressure sensor were threaded from the transmitter into the tunneling rod and pulled underneath the skin rostrally to the cranial cut.

The locations for the EEG screws were identified relative to landmarks on the skull. The position of the anterior EEG screw ($F_4$) was identified just anterior of the coronal suture, approximately 0.5cm to the right of midline. The position of the posterior EEG screw ($O_1$) was identified just anterior of the occipital ridge, approximately 0.5cm to the left of the midline. $F_4$ and $O_1$ refer to positions on the 10–20 EEG map [27]. The location of the intracranial pressure (ICP) sensor was identified over the right hemisphere between the anterior EEG electrode and the occipital ridge. To reduce the likelihood of subsequent tissue growth underneath the implant, circular regions of ~1cm diameter around these locations on the skull were sealed with three layers of copalite varnish (Temrex Corporation). The sites of instrumentation placement on the macaque cranium are illustrated in Fig 1.

Holes for the EEG screws were drilled using a 1.2mm surgical hand-operated drill. The self-tapping titanium EEG bone screws (Crist Instruments Co, Inc. 6-YXC-035) were fastened in place at a depth that would allow them to touch but not dimple the underlying dura. A 1mm-diameter, 2mm-deep hole had been drilled in the center of the internal hex-head of the screw prior to the surgery. The lead wires of the DSI transmitter were affixed to the EEG screws by means of a 1mm amphenol pin that was subsequently soldered to the EEG leads. After protruding portions of the amphenol pin and excess solder were clipped with pliers, the screws and pins were covered with dental acrylic to prevent damage and to preserve the integrity of the lead junctions. The access hole for the ICP transducer was drilled using a 2.5mm surgical drill. An electric drill with a dental drill-bit was then used to file down the ridge of the posterior aspect of the hole and to provide an approach parallel to the dura for the ICP sensor wire. A small plastic probe was inserted to carefully loosen the attachment of the dura to the skull. After removing the protective sleeve, the ICP sensor was carefully inserted into the space between dura and skull. The hole and the ICP apparatus were likewise sealed and covered with dental acrylic. Post-operative care consisted of 3 days' administration (PRN) of IM buprenorphine for analgesia and 5 days' administration (BID) of IM cefazolin for infection control. One out of 50 animals (including the 12 reported here) implanted with an EEG/ICP telemetry device developed complications post-surgery and was found deceased. A review of surgical methods used for that surgery did not reveal any deviations from procedure compared to surviving animals.

## Virus culture and aerosol challenge

Macaques were infected via aerosol exposure with VEEV strain IC INH9813 or EEEV V105 strains derived from cDNA clones (Sun et al., Manuscript submitted). Macaques were exposed to aerosols in a modified Class III biosafety cabinet in the Regional Biocontainment Laboratory (RBL) with methods pertaining to aerosol generation, aerosol exposure, respiratory

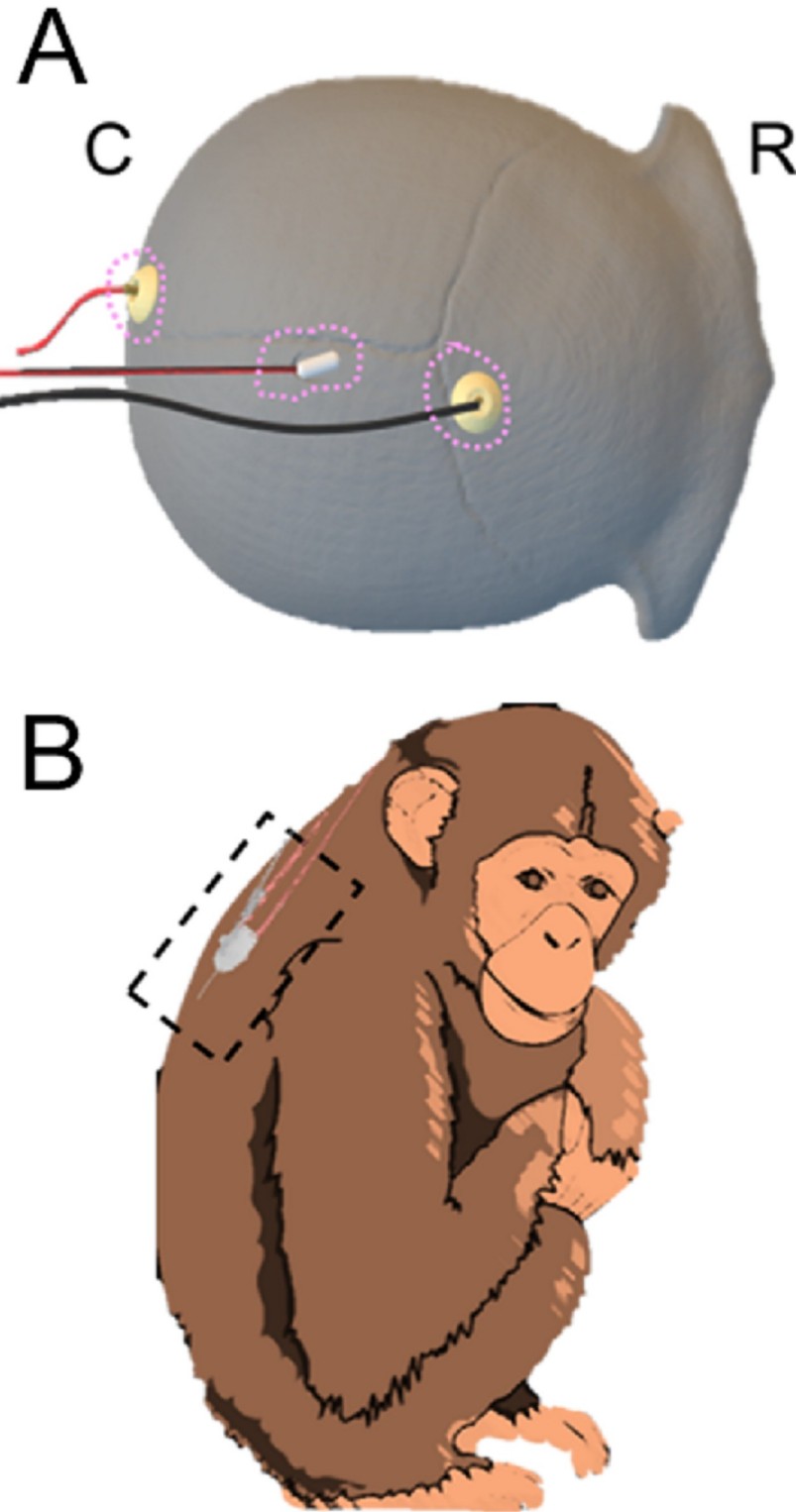

**Fig 1. EEG/ICP instrumentation placement sites.** A) EEG biopotential electrodes and ICP transducer dual modality, locations marked on macaque skull, top-down view. Anterior/cranial aspects face the right of the page, Posterior/caudal aspects face the left of the page; calvarial sections at top with "R" for rostral and "C" for caudal orientation. Pink outlines illustrate profile of dental acrylic cone used to encase sensors. B) Location of implanted telemetry transmitter over macaque scapulae. (CREATIVE COMMONS LICENCE Servier Medical Art by Servier is licensed under a Creative Commons Attribution 3.0 Unported License).

function determination, and inhaled dose determination as previously described (refer to Table 2 for doses) [14, 28–31]. Mock aerosols proceeded with all other conditions held equal except liquid medium supplied to the Aeroneb nebulizer consisted only of Opti-MEM® media with no virus in suspension.

## Data acquisition

The collection apparatus, located inside the RBL, consisted of a designated telemetry desktop computer outfitted with a communication link controller (CLC) supplied by DSI and signal collection from the implanted transmitters equipped with the Ponemah software package (v.5.20 SP8; DSI). The CLC was optimized for collecting signals from transceivers mounted on caging. All implanted transmitters contain a temperature sensor that samples core temperatures continuously, as well as a three-axis accelerometer designed to track body movement. As the macaques are singly housed within normal or purpose-designed biocontainment caging, closed circuit cameras (Axis Model No. M-1145) were positioned to continuously record macaque behavior. Daily review of video, neurological abnormalities, food and water intake, activity, appearance, stool/urine output, and aberrant clinical findings helped to reinforce clinical scores. Telemetry data was collected continuously for a baseline period of at least three days preceding aerosol challenge as well as for the days following aerosol exposure. The Ponemah software package provides a graphic user interface (GUI) of a unified, multichannel display of EEG and/or ICP data collected from the implant's sensor modalities. Raw EEG trace collected in this manner can be seen in S1 Fig. For analysis, data can be either read in raw form (.RAW file extension) or reviewed in common spreadsheet software with user-determined logging rates (i.e. 1/5/15 minute averages). Export of raw data to NeuroScore (v.3.0.0, DSI) facilitated batch processing of data from implanted transmitters to European Data Format (.edf), readable in MATLAB.

## EEG/ICP data analysis and statistical methods

EEG and ICP data were analyzed for each macaque using in-house MATLAB scripts building on routines from the MATLAB toolbox EEGLAB posted on GITHUB [32]. First, the 500Hz raw data was imported into MATLAB using the *edfread* function and converted to the EEGLAB data format. The data was then filtered using a 256-point noncausal digital low-pass finite input response (FIR) filter (*firws* function from EEGLAB toolbox, Blackman window, high-frequency cutoff 50Hz, transition bandwidth 21Hz). We then calculated a windowed fast Fourier transform (wFFT) using a cosine taper (400ms long flanks) for sliding 4s windows stepping in increments of 4s. After excluding 4s bins with EEG artifacts or excessive motion, the remaining spectra were then averaged within 15-minute long bins. We defined four distinct frequency bands (delta: [0.5 – 4Hz]; theta: [4 – 8Hz]; alpha: [8-12Hz]; beta: [12-30Hz]) and estimated the power in each band as the average power of all frequencies in the corresponding range. This resulted in four time-series that represented power in each enumerated band sampled at a rate of 1 sample per 15 minutes.

To visualize the circadian modulation of EEG power bands we created a 'circadian index.' To calculate the circadian index, we first normalized each of the power band time-series to a mean of 0 and a standard deviation of 1. The circadian index was then defined as the difference between the normalized delta and beta power. Since delta power peaks during slow-wave sleep and beta power during alert wakefulness, the circadian index should exhibit large positive values during night time and negative values during day time. Data were evaluated using Neuro-Score software packages, MATLAB 2018b (Mathworks), and GraphPad PRISM 8. For both individual animals as well as experimental group comparisons, t-tests and repeated measures

ANOVA tested for deviations from baseline. Both EEG and ICP data underwent frequency spectrum analysis through power spectral density plots. Linear regression and curve fitting were applied to ICP data.

## Results

We implanted 12 cynomolgus macaques with our multi-modal telemetry implants. Following at least 3 days of baseline data collection, the macaques were infected either with EEEV (6 macaques), VEEV (4 macaques) or mock (2 macaques). The data from the baseline period served as the basis for characterization of normal cynomolgus macaque EEG and ICP. Data from the two mock infected macaques served as the basis to evaluate the stability and consistency of the relevant measures in the absence of an infection. Data from the experimental macaques served as the basis to identify pathological changes that are specific to the two diseases and/or predictive of the outcome of the infection.

### Electroencephalography

To extract interpretable data from the raw EEG traces, they were analyzed in the frequency domain. Spectral analysis of raw EEG traces is important, because different frequency bands have been linked to different aspects of brain function. For example, slow-wave frequencies such as those encompassing the delta band, play a role in the regulation of and/or reflect the status of homeostasis maintenance, while higher frequency waves are associated with higher brain functions as well as muscular coordination and feedback [33–38]. In primates, including humans and macaques, sleep can be characterized into non-rapid eye movement sleep (NREM) and rapid eye movement sleep (REM); NREM sleep is correlated with slow wave EEG activity, as encompassed by the delta wave band [39–41]. Delta activity, as such, peaks during slow wave sleep, while activity in the higher frequency wave bands, such as the beta wave band, predominates during alert states [41–43]. This feature of the EEG frequency spectrum makes it relevant to the qualification and quantification of circadian cycles.

Initially, a frequency spectrum analysis was done to assess the normal pattern for EEG data across the four power bands. The average EEG power spectrum of a representative example macaque is shown in Fig 2A. The approximately linear decline of EEG power with frequency (when plotted on a log-log scale) is expected and suggests that the leads were implanted correctly and record physiologically plausible EEG activity. While there is some variability in the daily averages, there is no systematic drift over the entire recording period. This speaks to the utility of the spectral analysis as a tool to identify pathological changes of brain function. Fig 2B shows the 12 average spectra for all macaques during the baseline period. Note the high similarity of power-spectra across the entire population. This suggests that the surgical procedure was successful at providing replicable electrode positions and electrical contacts for all macaques. Note that only three macaques show a peak at specific frequencies at baseline. The presence of such idiosyncrasies would suggest that pre-exposure baseline data, if available, can further enhance ability to detect pathological changes by serving as its own control [3, 12, 14].

To test for the presence of circadian EEG rhythms, we averaged EEG spectra separately for all twelve macaques during the baseline period over the light and dark conditions of the animal holding room in (Fig 3A). This initial analysis suggested a high degree of circadian variability in the beta band (p<0.0001). To better understand the temporal dynamics of these circadian variations, we extracted time-series data for all four EEG power bands in bins of 15 minutes. Fig 3B documents the averaged circadian modulation of these power bands in a single mock-infected macaque, the same macaque as in Fig 2A. As expected, beta activity coincides roughly with activity and body temperature and is highest during daytime while delta peaks at night

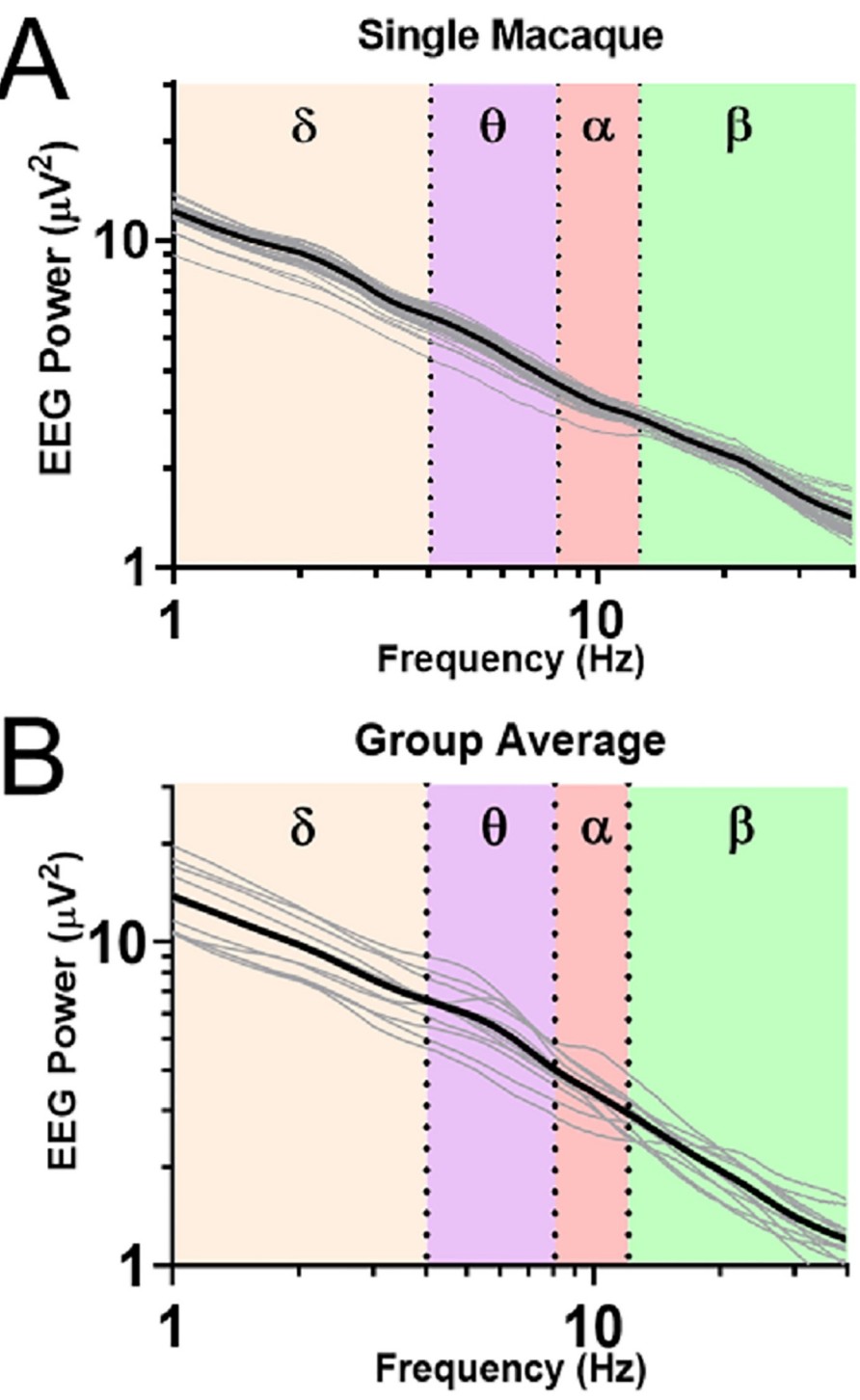

**Fig 2. EEG baseline frequency spectral analysis in normal macaques.** A) A power spectral density (PSD) plot from the entire time course of a mock-infected macaque. Highlighted by different background colors are the delta (0-4Hz), theta (4-8Hz), alpha (8-12Hz), and beta (12-30Hz) bands. Daily averages are indicated by thin gray lines. B) Pooled baseline data from all macaques in the study. Individual macaques are indicated by thin gray lines.

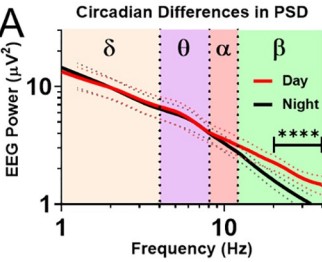
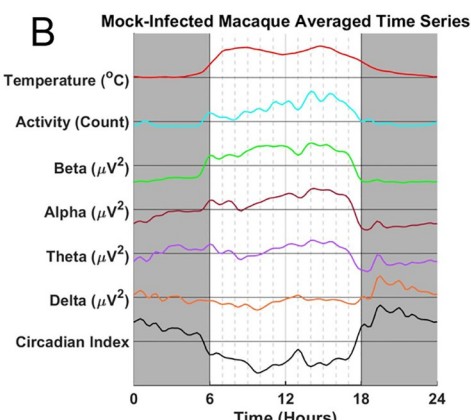

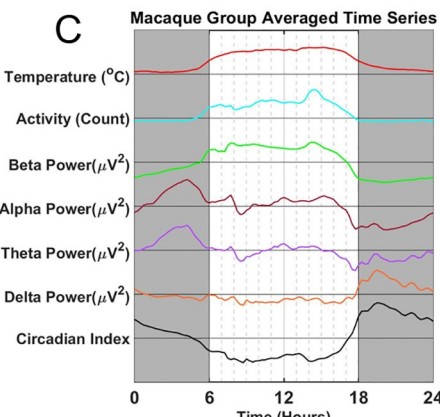

**Fig 3. EEG circadian variability.** A) Population spectral density plots separated by day (red) and night (black). Note the increased beta-band activity during the day (p<0.0001) B) Circadian modulation of alpha, beta, theta, and delta power for one mock-infected macaque (M115-16). Gray boxes denote night, or dark periods, i.e. no lighting in the holding room. C) Pre-infection population average of all days' circadian modulation. Beta-band activity is elevated throughout the day; Delta-band activity peaks in the first half of the night; alpha- and theta- band activity peaks in the second half of the night.

when activity and body temperature are lowest. Similar patterns are seen for pooled, averaged baseline data for all macaques in the cohort across a similar 24-hour span is shown in Fig 3C. At the group level, a number of details of the circadian pattern emerge. Specifically, the group data show a transition during the night from a delta peak in the first half of the night (1800 to 2400) to a theta and an alpha peak in the second half of the night (0000 to 0600). It is also interesting to note that beta power starts ramping up a few hours before the day (facility lights-on). These transitions of the EEG power spectrum during the night suggest diurnal arousal because they cannot be explained by motion artifacts, as activity stays flat during the entire night.

The natural circadian variation of the EEG power bands is largest for the beta band, and is also present in the other power bands. The detection of pathological changes thereby requires discrimination of anomalous peaks and troughs against a constantly varying background. Such discrimination of pathological changes from natural circadian variation depends not only on the power bands' absolute magnitude, but on the reliability of the time series data on a day-by-day basis. Fig 4A illustrates daily circadian rhythms over an 18-day period in an example mock-infected macaque. Note that the circadian rhythms were stable over the course of the entire experiment. The high predictability of these circadian rhythms should facilitate the

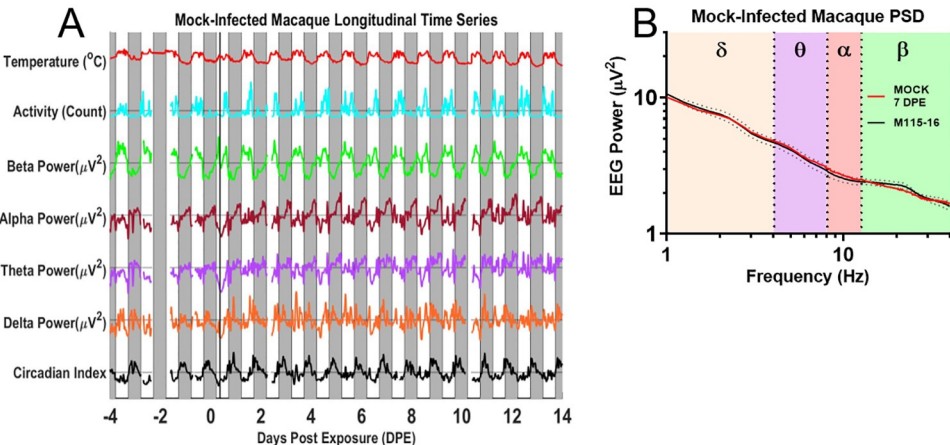

**Fig 4. EEG processed time series in the uninfected macaque.** A) EEG time course and accompanying B) PSD plots from Mock-infected macaque (M115-16). Solid magenta bar on Day 0 indicates time of aerosol exposure, gray boxes indicate night.

detection of pathologic changes. The data in the figure represent the envelope surrounding the filtered EEG data, and convey less noisy traces while preserving trends of increase and decrease in the wave band magnitude. The PSD plot (Fig 4B) demonstrated no significant deviations between pre-exposure and post-exposure spectra in this mock-infected animal.

To confirm that pathological changes can be observed on the baseline of predictable circadian rhythms, we turned to the two experimental groups infected with EEEV and VEEV, respectively. Experimentally infected macaques manifested a number of neurological signs of disease expected to accompany alphavirus encephalitis. EEEV-infected macaques with severe disease typically manifest a severe febrile course at 3 DPE, and neurological signs of disease included head pressing, involuntary tremor and seizures, which persisted until humane study endpoint and euthanasia [3]. VEEV-infected macaques manifest a biphasic febrile disease course with the first fever peak occurring from 0.5 to 2DPE, and the second peak occurring between 2.5 to 8 DPE, as previously observed [12].

Fig 5A shows data from one macaque that developed severe disease after EEEV infection. Initially, the time series analysis shows a regular and robust circadian modulation much like the mock-infected macaques. However, starting on 3DPE, a number of increasingly dramatic changes are evident that diverge from the normal circadian cycle. The first abnormality consists of an increase of the alpha power band on 3 DPE. This alpha peak coincides with fever onset. In the subsequent night, abnormalities start developing in the delta power band. While the early night delta peak still emerges, it is cut short and followed by an unusual drop in delta power that extends into 4DPE and reaches a dramatic trough on during the early morning (before lights-on) of 5DPE, contemporaneous with peak fever. Concurrently, power drops in all other frequency bands; however, the drop in delta power is particularly noteworthy since delta power in this macaque typically shows a clearly-delineated early night-time peak. The morning of 6DPE brings another dramatic change that manifests as an increase of power across all bands. Both the uniform drop of power on 5DPE as well as the uniform increase of power on 6DPE are highly unusual, given the normally inverse relationship of delta power on the one hand and theta-, alpha-, and beta- power on the other. The pattern of uniformly increased power holds for the entirety of 6DPE as the fever begins its terminal decline. The unusual coupling of the power bands can also be observed as a decrease in the amplitude of the circadian index on 5-6DPE. Throughout the time course, dips in temperature from the

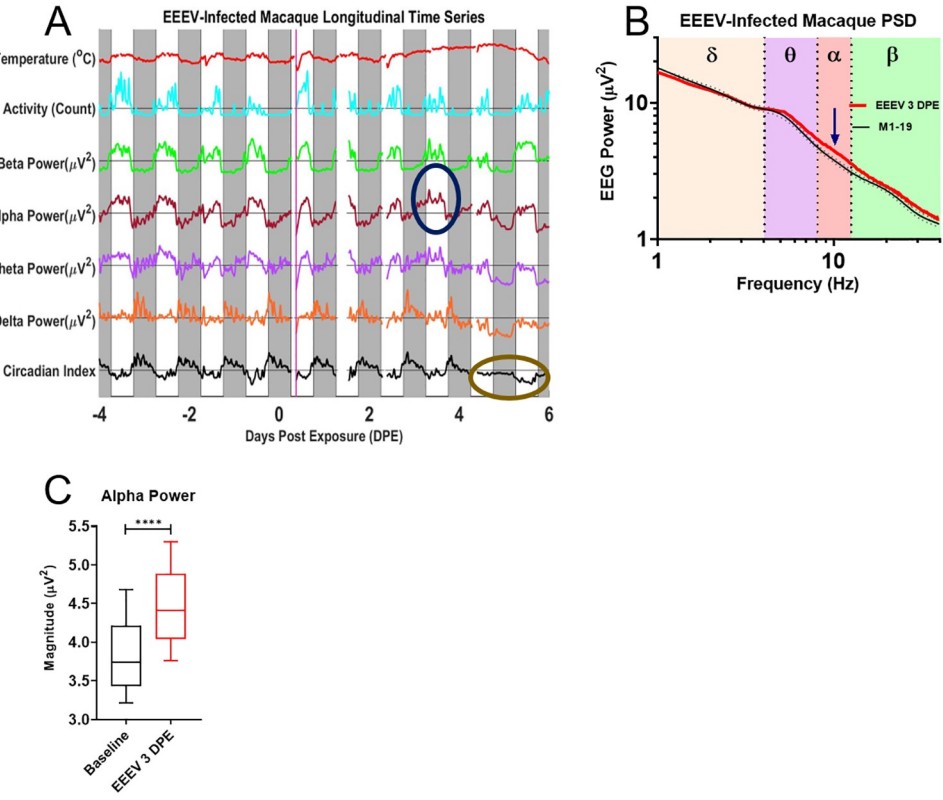

**Fig 5. EEG processed time series in the EEEV-infected macaque.** A) EEG time course and accompanying B) PSD plot from EEEV-infected macaque (M1-19), dotted lines represent upper and lower bounds as standard deviation. Blue circle indicates increases in alpha activity on 3DPE, visualized on the PSD plot (red trace) emphasized with corresponding blue arrow in comparison to baseline spectra for that macaque, note also taupe ellipse marking suppression of sleep index from 4–5 DPE. C) T-test of alpha power between 3DPE and baseline data demonstrate statistically significant increase (p<0.0001). Febrile period for severe EEEV begins at 3DPE and ends at euthanasia (~5-6DPE). The solid magenta bar on Day 0 indicates time of aerosol exposure, while gray boxes indicate night.

administration of anesthesia for blood draws can be seen in the temperature trace. The macaque was determined to be moribund and euthanized on day 6. A PSD plot confirmed the increase in alpha power band magnitude on day 3 was significant (Fig 5B and 5C).

VEEV-infected macaques manifest a biphasic fever following infection, shown in Fig 6A alongside activity and power band traces from one macaque that developed disease following VEEV infection. Note also slight dips every other day in the temperature trace due to anesthesia administered for blood draws. Initially, the time series analysis also shows a regular and robust circadian modulation much like the mock-infected macaques. However, on 7 DPE, there was a reduction in both beta power and delta power, with the latter finding being part of a trend ranging from 5–10 DPE, and a suppressed circadian index. These findings were also visible in the PSD plots and subsequent statistical analyses (Fig 6B, 6C and 6D). It is notable that both of these changes are seen during the second febrile period, which has been hypothesized to be a period in which the virus penetrates the CNS [44]. Note the massive delta peak in night 12 post-infection. This may correspond to a rebound of slow-wave sleep after several days without any discernable slow wave sleep activity during the night. Across the population of macaques exposed to alphaviruses EEEV or VEEV, the blunting of the circadian rhythms is the most common pathological change. Its duration for VEEV-infected macaques is typically longer than for the EEEV-infected macaques, though the heightened severity of disease in EEEV-infected macaques, whose

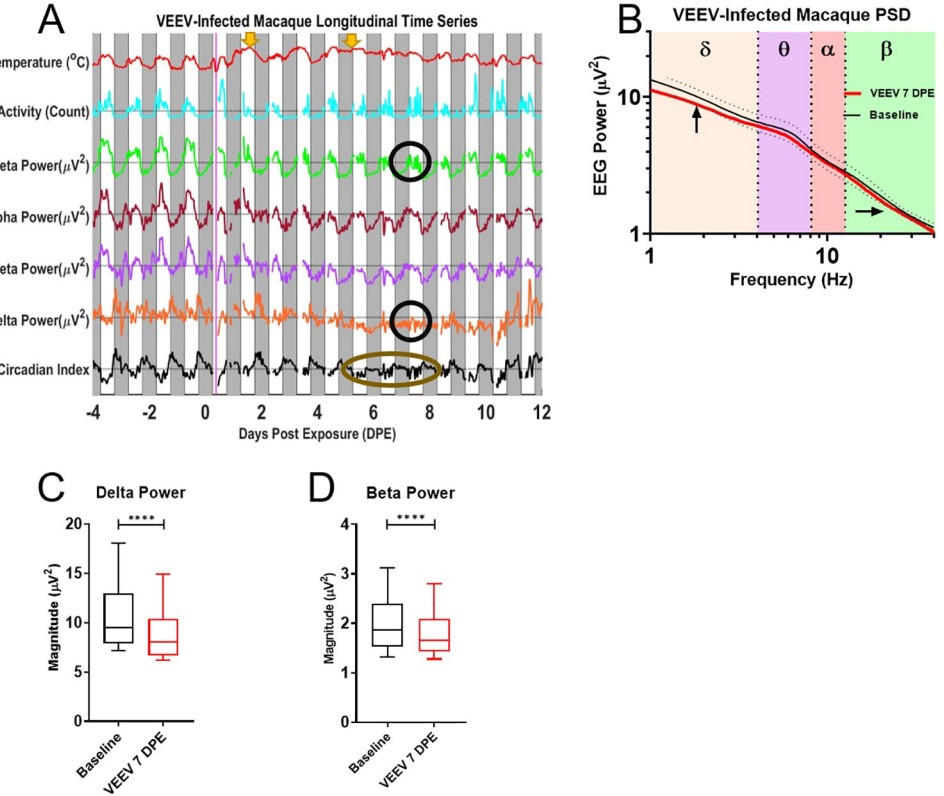

**Fig 6. EEG processed time series in the VEEV-infected macaque.** A) EEG time course and accompanying B) PSD plot from VEEV-infected macaque (M115-18); dotted lines represent one standard deviation as upper and lower bounds. Black circles indicate decreased delta power and beta power on 7DPE, visualized on the PSD plot with corresponding arrows in comparison to baseline spectra for that macaque. Note also taupe ellipse marking suppression of sleep index from 5-8DPE. C) T-test of delta power and D) beta power between 7DPE and baseline data demonstrate statistically significant decreases (p<0.0001). The febrile period for VEEV is biphasic with two fever periods from 0.5–2 DPE and from 2.5–8 DPE; periods marked by yellow arrows. Solid magenta bar on Day 0 indicates time of aerosol exposure, gray boxes indicate night.

neurological signs of disease warrant euthanasia during the febrile period, may mask a similar effect. Additionally, the EEG spectra of macaques infected by VEEV demonstrated fewer statistically significant deviations from baseline data than EEEV-infected macaques.

## Intracranial pressure

Increased ICP during alphavirus encephalitis has been reported in humans [45–49]. Increases of ICP are typically a delayed and indirect result of brain injury, which ultimately leads to inflammation and swelling; in addition, inflammation and swelling of deep brain structures can affect venous drainage and CSF circulation, thus leading to a secondary rise in ICP. To measure ICP in the macaques, we inserted a standard blood-pressure sensor in between the dura and the skull. Because blood pressure is much higher than intracranial pressure, blood-pressure sensors are tuned to a different range of pressures. Hence, we first wanted to verify that we could detect meaningful changes of ICP using this blood-pressure sensor. It is well known that ICP fluctuates with individual heartbeats, known as the cardiac pulsation of ICP. If the blood pressure sensor were indeed sensitive to physiological variations of ICP, the collected telemetry data would presumably register an ICP pulsation corresponding to the cardiac pulsation, within the range of physiological values of the macaque heart rate (here defined as

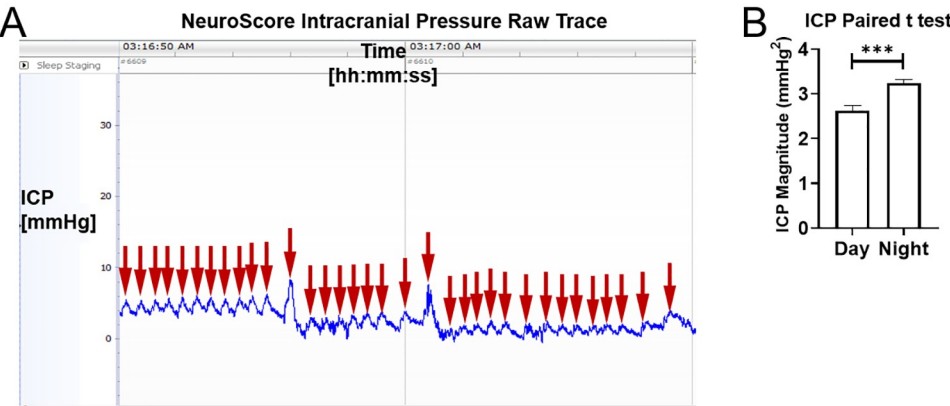

**Fig 7. Intracranial pressure data and circadian variation.** A) The raw ICP trace in NeuroScore from a single example macaque exhibits individual heartbeat pulsations (red arrows). B) ICP exhibits significant diurnal differences (p<0.01), from the day and night averaged ICP values of a mock-infected macaque.

between 1.5 and 3.5 Hz, corresponding to 90 to 210 bpm). Fig 7A shows a segment of raw ICP trace for one example macaque. Note the clearly identifiable cardiac pulsations. This demonstrates that the blood pressure sensors of the M11 and L11 implants were exquisitely sensitive to the cardiac pulsation of ICP within the expected physiological range. In humans, normal values of intracranial pressure range from 0–15 mmHg, varying according to postural changes; though circadian variation in ICP has been documented, such variation may be due in large part to expected changes in postural preferences during awake and asleep states [50, 51]. Though few studies have comprehensively studied the subject, reports in the literature have demonstrated that ICP values in adult cynomolgus macaques fall into a similar range of normal pressures. A combination of our observations, the activity traces seen in Figs 3B, 3C and 4A, and documented reports of postural preferences in awake and asleep macaques suggest the possibility of diurnal variation of ICP in cynomolgus macaques due to associated variation in posture and movement [52, 53]. Deviation from a circadian pattern can provide an additional piece of evidence for the onset of a pathological process in a monitored macaque. Fig 7B demonstrates the circadian variation of the intensity of the ICP. Comparisons of daytime and nighttime values of the data obtained through frequency spectrum analysis indicated that measured pressures tended to display increased values at night (p<0.01). This finding, a group measure, can reflect the general preference for a seated, head-down posture during the night [53]. Alongside the understanding that supine or seated postures can produce increased ICP relative to standing, upright postures, the output of increased nighttime intracranial pressure in this set of macaques represented a fairly reasonable outcome [54].

We next focused on the stability of the ICP measurements over the measurement periods of many days in mock-infected macaques. Fig 8A shows an example ICP trace superimposed upon a temperature trace in such a subject. ICP is mostly stable over the course of multiple days despite some outlier pressure values observed during anesthesia administered for blood draws or aerosol exposure. The relative stability of the ICP measurements over the course of many days should facilitate the detection of pathologic changes after viral infection. We next tested whether we could indeed observe pathological increases of ICP associated with acute viral encephalitis. Fig 8B shows the ICP trace of an example macaque that was infected with EEEV and developed a severe encephalitis. The ICP began to rise at about the time the fever peaked at 4 DPE. Note that ICP remained at peak levels well after the fever peaked and until the macaque was deemed moribund and qualified for the humane study endpoint. Fig 8C

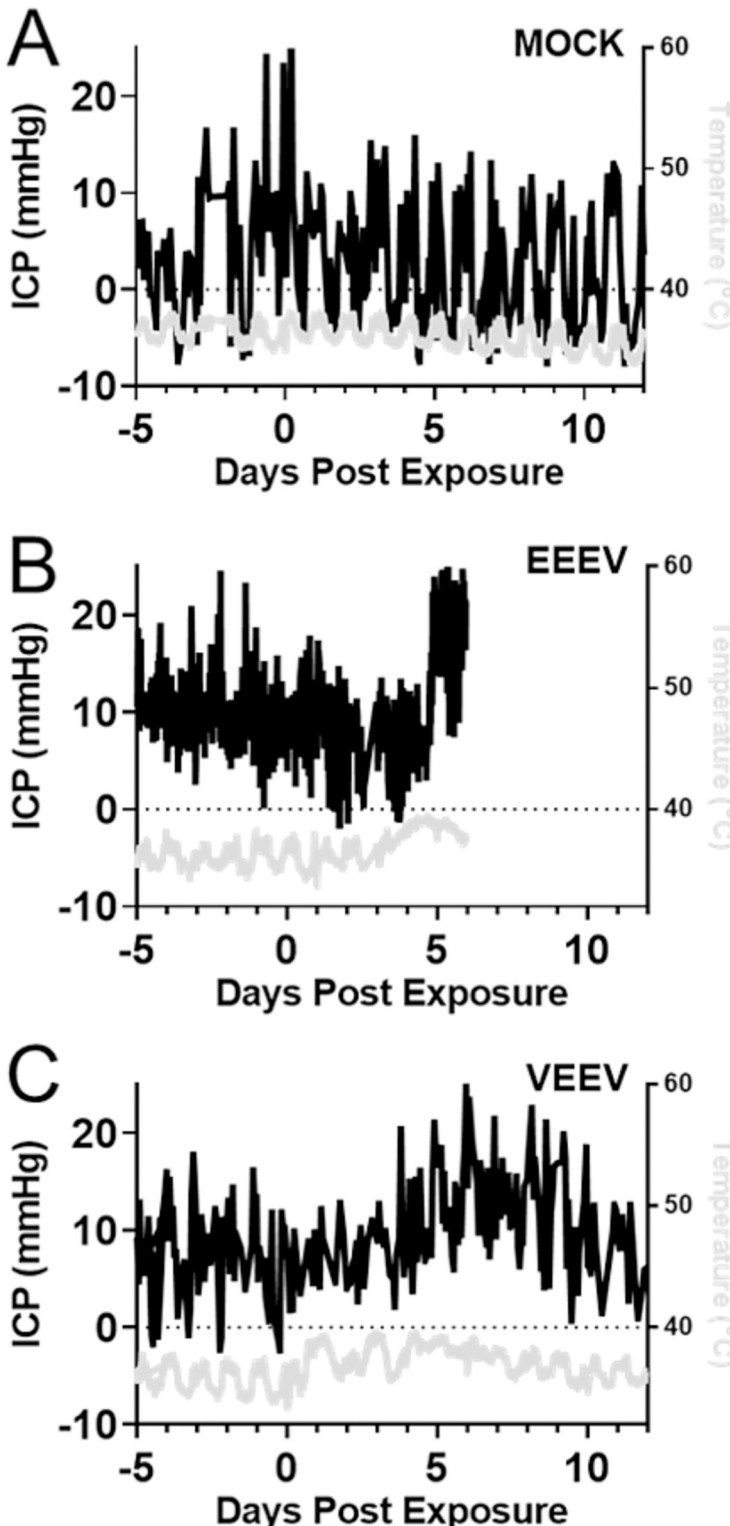

**Fig 8. Raised ICP in alphavirus encephalitis.** A) ICP time series in a mock-infected macaque, superimposed on temperature curve. B) EEEV-infected macaques typically exhibit the largest ICP increases during the febrile period; an example ICP trace from a representative macaque (3–19) is superimposed upon temperature helps to illustrate the rise in ICP in relation to the fever curve. C) Similarly, in a representative VEEV-infected macaque (115–18), intracranial pressure rises after the onset of a characteristic biphasic fever. Temperature curves comprise gray traces.

shows the ICP trace from a VEEV-infected macaque with a mild disease outcome. An increase in ICP was observed around the time of the second febrile period. The rise in ICP took place at a slower rate compared to the EEEV-infected macaque, and the ICP eventually returned to baseline at approximately 12 DPE. What is notable in both EEEV and VEEV is that the increase in ICP is seen after the fever has peaked. After becoming sick, infected macaques spent more time in a hunched, sitting posture, and sometimes pressed their heads against the sides of cages, a sign of neurological disease.

ICP measurements can be contaminated by "zero-drift," a slow artificial change in the measurements caused by a gradual de-calibration of the sensor or device [55]. The fast changes of ICP observed in the infected animals are not consistent with such gradual de-calibration. However, we wanted to confirm the increases in ICP during the periods of clinical disease using a secondary measure putatively unaffected by potential zero-drift. To that aim, we took advantage of the well-known effect that cardiac ICP pulsations are more intense during conditions of increased ICP. In other words, we wanted to test whether the amplitudes of cardiac ICP pulsations were elevated during periods of increased ICP. This would be a strong independent confirmation that the observed increases of ICP during acute encephalitis are real, and not caused by zero-drift. Representative data from one example animal shows that the amplitude of cardiac pulsations indeed increase in line with mean ICP during encephalitic disease (Fig 9A). Moreover, as expected with acute viral encephalitis, the heart rate increased in infected animals during the febrile period as previously documented [3, 14]. This shift was also visualizable in the power spectral density plots (Fig 9A), providing further evidence for the functionality of the ICP implant modality. The frequency spectra of data collected for a total of 33 days (Fig 9B) in a mock-infected macaque demonstrated no statistically significant changes in the pressures taken continuously before exposure to a control media aerosol and through 27 DPE. Furthermore, we show that the amplitude of cardiac pulsations and mean daily ICP are linearly correlated (Fig 9C); the se analyses were performed with the post-exposure data of mock-infected macaques as an invariant control alongside temporally-matched EEEV post-exposure data, before the humane study endpoint. Taken together, these additional analyses provide strong evidence that the observed increase of ICP during acute encephalitis are genuine.

## Discussion

Nonhuman primates, including the cynomolgus macaque (*Macaca fascicularis*) have served as model organisms for equine encephalitis viruses since the discovery of these viruses in the 1930s [56, 57]. Aerosol infection of macaques with equine encephalitis viruses reproduces central nervous system lesions similar to those in human cases [3, 12, 58, 59]. Here we successfully implanted EEG/ICP telemetry devices that were designed to provide non-invasive and continuous measures of several symptom dimensions in a macaque model of viral encephalitis. The first part of the report focused on detailing the implantation of the devices and analysis methods in order to allow others to replicate and extend our technical efforts. The second part presented a number of intriguing findings that showcase the utility of the implants to capture the interplay between different symptom dimensions and the sometimes brisk transitions between different stages of the acute encephalitis. These findings encourage the use of multi-modal continuous telemetry in subsequent alphavirus encephalitis studies for the evaluation of medical countermeasure efficacy.

### Electroencephalography

EEG is a biopotential signal that arises from synchronous post-synaptic potentials in large groups of neurons. Key aspects of EEG are conserved across many species and thus applicable

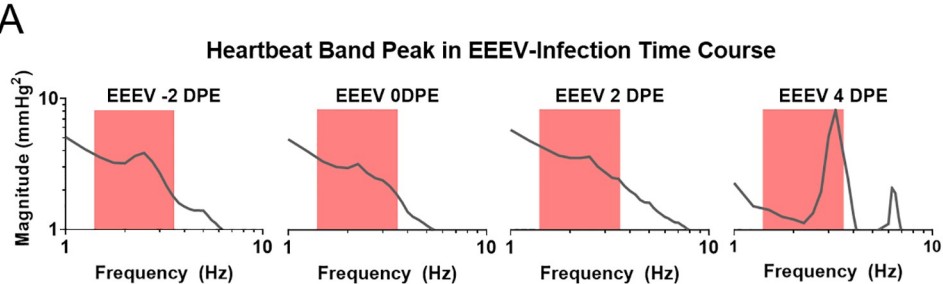

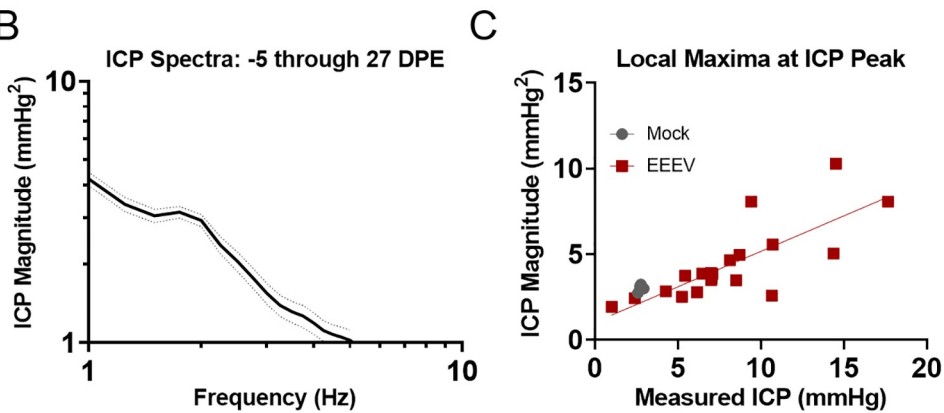

**Fig 9. ICP measurement stability and correlation to time series grand averages.** A) ICP power spectral density plots from representative EEEV-infected macaque (M3-19) demonstrate not only a rise in pressure intensity visualized as local maxima, but also a rise in heart rate as the maxima shift right from baseline to the febrile period between -2DPE and 4DPE (red box marks a truncated physiological range of heart rate from 1.5–3.5Hz). B) B) No statistically significant changes in ICP occur over the study course from -5 days through 27 DPE in the frequency spectrum. C) The local maximum magnitude scales proportionally with aggregate daily ICP values measured from the raw ICP trace.

to macaques and humans. The continuous long-term monitoring in mock-infected animals allowed us to characterize normal variations across the time of day, reflecting normally present biological rhythms in different frequency ranges of brain activity. This data established a baseline against which to evaluate pathological brain function caused by infection with an encephalitic alphavirus. Pre-infection, all frequency bands exhibited circadian variation. As may be expected, the slow-frequency wavebands delta and theta, exhibited higher magnitudes during the night and lower magnitudes during the day, while the high-frequency wavebands alpha and beta exhibit lower magnitudes at night and higher magnitudes during the day. Subtle differences in timing make themselves manifest; for instance, the delta magnitude peaks early during the night, whereas theta and alpha magnitude tend to peak after midnight. The beta magnitude also begins ramping up after midnight, with a sharp uptick after lights-on. The data from each PSD plot are lognormally distributed rather than normally distributed, and differences between macaques' baseline data suggest that group-wise statistical analysis of EEG data can be strengthened by normalizing each macaque's data to its own baseline period.

The utility of monitoring EEG was demonstrated by the differences seen between mock-infected and alphavirus-infected macaques; infection of the macaques with EEEV and VEEV in this cohort provided a tangible contrast in EEG to the mock-infected macaques. Infected macaques displayed a variety of EEG abnormalities during the post exposure period, namely

differences in the delta, alpha, and/or beta band. In both EEEV and VEEV, a prominent suppression and loss of circadian rhythm in delta magnitude is seen in the late febrile period (5–10 DPE). Delta bands returned to normal in VEEV-infected macaques after 10 DPE while EEEV-infected macaques succumbed to infection so we believe this loss of delta magnitude is reflective of CNS disease but does not predict outcome. Further study with a more intensive group analysis is warranted to clarify this relationship.

Multiple reports in the literature comment on the loss of circadian variability in various segments of the EEG frequency spectrum during viral infection, with various empirical associations outlined that implicate both systemic and neurogenic origins of circadian disruption [18–21, 60–63]. Acute phase reactants such as tumor necrosis factor (TNF-α) have been tied to slow-wave EEG activity, including the delta power band; TNF-α is thought to directly or indirectly regulate somnolence [21]. Endogenous fluctuations of inflammatory or immune species in response to infection may thereby produce changes detectable by EEG. Our findings hint at an inflammatory neurogenic origin of disrupted circadian rhythmicity. For instance, the first febrile peak in VEEV is associated with virus dissemination through the lymphatic system, while the second febrile period is proposed to be associated with infection of the CNS [12, 44]. Circadian disruptions of EEG in VEEV infection occur largely during the second febrile peak, suggesting that infection of the brain parenchyma and/or the subsequent immune response cascade precipitates the disruption. Holistically, the choice to present findings pertaining to the circadian index arose out of the practical consideration to illustrate the longitudinal nature of brain pathology in the context of alphavirus encephalitis. This consideration leverages the continuous nature of circadian index construction over the assessment of seizures, which did not occur in all macaques with alphavirus encephalitis and which presented nonuniformly and episodically. Evaluating seizure activity in cynomolgus macaques would not be useful in assessing efficacy of medical countermeasures against alphavirus encephalitis. With respect to the infectious origin of alphavirus encephalitis, the biomolecular milieu that facilitates encephalitis also affects, as described above, the regulation and dysregulation of the circadian variation of EEG, captured by the circadian index. Our findings suggest that the assessment of the circadian index can provide a valuable research tool for the determination of the onset, resolution, and/or mitigation of disease in the context of the evaluation of medical countermeasures against acute viral encephalitides.

## Intracranial pressure

Measurements of ICP provide a proximate measurement of the onset and resolution of the encephalitic disease state. Our results show that we can successfully measure ICP in the macaque using a blood-pressure monitor. As a proof of concept, we demonstrated that the blood-pressure monitor can pick up physiological changes of ICP, such as cardiac ICP pulsations, and that these physiological signals did not degrade even after several weeks' worth of data collection. Most importantly, our results show the EEG/ICP implants can detect the expected increases of intracranial pressure during the acute phase of encephalitis. Increase of ICP was found to lag behind the onset of fever and instead emerge around the time of the peak of the fever. For VEEV, increase of ICP was tied to the second phase of the fever that is believed to reflect the infection of the brain.

## Study limitations

The data presented here provides a number of intriguing results that are based on our ability to continuously monitor EEG, ICP and temperature. Additional work is currently under way to test whether these findings hold up in a study with larger sample size (manuscripts in

preparation). In particular, the temporal relationship between fever and the emergence of EEG and ICP anomalies merits more in-depth attention. The present report detailing the implantation of the devices and analysis methods is important for allowing others to replicate or extend our technical efforts both for continued study and for generalizability of this model for other models of encephalitis.

The EEG/ICP implant used for this study permitted only one single bi-polar EEG recording. Standard EEG caps in humans typically contain 32 to 64 channels that are spaced approximately evenly across the scalp. Such dense spatial sampling enables the use of source reconstruction techniques that can map certain EEG events to specific brain regions. In the current context, a larger number of electrodes might similarly enable the mapping of pathological changes to specific brain regions, particularly with regard to the algorithmic detection of seizures. Such analyses would be particularly informative in combination with post-mortem pathology in different brain regions. Because of the two-lead limitation in the detection of seizures with a high false positive rate by algorithmic detection, the task fell to human observation and confirmation of seizure activity either through incidental observation or through in-depth review of hundreds of hours of video recordings.

EEG electrodes are easily affected by electric sources other than the brain, e.g., saccadic eye movements, blinking, and chewing [64, 65]. Our pre-processing steps were aimed at excluding epochs with obvious artifacts, but it was impossible to detect and exclude many of the more subtle artifacts. In future studies, artifact detection can be improved by using multi-electrode EEG arrays and using independent component analysis to identify artifacts based on their spatial distribution and temporal properties. Nevertheless, we are confident that many of the reported EEG abnormalities reflect abnormal brain function in the disease rather than altered artifacts from non-cerebral sources (e.g., less chewing in sick animals).

The current study used a blood-pressure sensor to measure ICP. While the sensor clearly was able to measure physiological changes of ICP, it is plausible that a dedicated ICP pressure sensor that is designed to have maximal sensitivity in the range of ICP would have provided less noisy data. Furthermore, it is known that ICP can be affected by body posture. Thus, it is plausible that accounting for body posture using an automated posture detection mechanism could further improve the signal-to-noise of the ICP measurements.

## Generalizability and translation

The augmentation of this well-established macaque model holds utility towards the development of improved vaccines. The macaques that exhibited courses of EEEV or VEEV disease displayed abnormal EEG and ICP measurements. Observed seizures occurred only in EEEV-infected macaques and typically comprised tonic-clonic seizures with generalized shaking and immobility for approximately 3–5 seconds' duration and a post-ictal period of approximately 5–10 seconds. These seizures were observed with greatest frequency at 4–6 DPE in EEEV-infected macaques. Seizures tended to occur on days with abnormal EEG. However, it was not possible to reliably identify seizure activity in the EEG. This may be due in part to the spatial limitations of our two-lead EEG system.

We conclude that the analysis of EEG spectra and the derived sleep indices may portend significant findings in higher-powered studies of acute viral encephalitic diseases in the future. The methods detailed in this work may prove generalizable to the investigation of other encephalitic disease states. The finding of accurately documented, raised intracranial pressure remains of particular interest in better characterizing encephalitic diseases of infectious origin. These findings in telemetered macaques hold the potential to track acute viral encephalitis

disease courses for the testing and evaluation of vaccine candidates to mitigate or prevent human disease.

## Supporting information

**S1 Fig. EEG data collection.** Raw data collected from the dedicated telemetry computer equipped with Ponemah software package provides a real-time, continuous display of EEG data, y-axis in units of mV. These data are downloaded to the NeuroScore software package and exported in .edf file extension format for analysis in MATLAB for frequency decomposition and reconstitution into delta, theta, alpha, and beta time series traces.
(TIF)

**S1 Data.**
(M)

**S2 Data.**
(M)

**S3 Data.**
(M)

## Acknowledgments

Special thanks to members of the Reed, Hartman and Klimstra labs at the University of Pittsburgh, especially Stacey Barrick and Theron Gilliland, Jr., and members of the Division of Laboratory Animal Resources, and Kate Gurnsey.

## Author Contributions

**Conceptualization:** Douglas S. Reed, Tobias Teichert.

**Data curation:** Henry Ma, Jeneveve D. Lundy, Emily L. Cottle, Katherine J. O'Malley, Douglas S. Reed, Tobias Teichert.

**Formal analysis:** Henry Ma, Douglas S. Reed, Tobias Teichert.

**Funding acquisition:** William B. Klimstra, Amy L. Hartman, Douglas S. Reed.

**Investigation:** Henry Ma, Jeneveve D. Lundy, Emily L. Cottle, Katherine J. O'Malley, Douglas S. Reed, Tobias Teichert.

**Methodology:** Henry Ma, Douglas S. Reed, Tobias Teichert.

**Project administration:** Anita M. Trichel, William B. Klimstra, Amy L. Hartman, Douglas S. Reed, Tobias Teichert.

**Resources:** Anita M. Trichel, William B. Klimstra, Amy L. Hartman, Douglas S. Reed, Tobias Teichert.

**Software:** Henry Ma, Douglas S. Reed, Tobias Teichert.

**Supervision:** William B. Klimstra, Amy L. Hartman, Douglas S. Reed, Tobias Teichert.

**Validation:** Henry Ma, Douglas S. Reed, Tobias Teichert.

**Visualization:** Henry Ma, Douglas S. Reed, Tobias Teichert.

**Writing – original draft:** Henry Ma, Douglas S. Reed, Tobias Teichert.

**Writing – review & editing:** Henry Ma, Anita M. Trichel, William B. Klimstra, Amy L. Hartman, Douglas S. Reed, Tobias Teichert.

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
