## [Decision Letter · Decision Letter 0]

15 May 2020

PONE-D-20-10011

Applications of minimally invasive multimodal telemetry for continuous monitoring of brain function and intracranial pressure in macaques with acute viral encephalitis.

PLOS ONE

Dear Dr Teichert,

Thank you for submitting your manuscript to PLOS ONE. After careful consideration, we feel that it has merit but does not fully meet PLOS ONE’s publication criteria as it currently stands. Therefore, we invite you to submit a revised version of the manuscript that addresses the points raised during the review process.

This manuscript is well written and scientifically sound. Reviewers have raised a few minor issues about the paper. Please consider reviewers’ recommendations and resubmit your work for publication after revision. I wish you the best.    

We would appreciate receiving your revised manuscript by Jun 29 2020 11:59PM. To enhance the reproducibility of your results, we recommend that if applicable you deposit your laboratory protocols in protocols.io, where a protocol can be assigned its own identifier (DOI) such that it can be cited independently in the future. For instructions see: http://journals.plos.org/plosone/s/submission-guidelines#loc-laboratory-protocols

We look forward to receiving your revised manuscript.

Kind regards,

Negin P. Martin, Ph.D.

Academic Editor

PLOS ONE

Journal Requirements:

3. In order to comply with PLOS ONE's guidelines for non-human primate experiments (http://journals.plos.org/plosone/s/submission-guidelines#loc-non-human-primates), please indicate how often animal care staff monitored the health and well-being of the animals and specify the disposition of animals at the end of the study (e.g. euthanasia, returned to home colony, etc.). If animals were euthanized following the study, please provide the method of sacrifice.

Additional Editor Comments (if provided):

Reviewers' comments:

Reviewer's Responses to Questions

**Comments to the Author**

1. Is the manuscript technically sound, and do the data support the conclusions?

Reviewer #1: Yes

Reviewer #2: Yes

2. Has the statistical analysis been performed appropriately and rigorously? 

Reviewer #1: Yes

Reviewer #2: Yes

3. Have the authors made all data underlying the findings in their manuscript fully available?

Reviewer #1: Yes

Reviewer #2: Yes

4. Is the manuscript presented in an intelligible fashion and written in standard English?

Reviewer #1: Yes

Reviewer #2: Yes

5. Review Comments to the Author

Reviewer #1: First of all, the authors did an excellent job in describing the procedures in details and demonstrating the potential utility of the macaque model for VEEV/EEEV infection. A primate model for this rare yet highly morbid encephalitis is indeed important, and objective quantification of data is necessary for reproducibility and generalizability of findings.

The use of ICP monitor, particularly in the extradural location, over several days without significant data disintegration is impressive. In humans, such a monitor is typically limited to a few days of use, after which data become unreliable. The authors did a great job analyzing the acquired data for a potential zero-drift, and showed that the acquired ICP tracing is likely "real" rather than artifactual. It would be nice to see integrated EEG and ICP data in infected animals to understand the time course of changes in acquired data relative to timing of clinical changes pertinent to infection (eg. vitals such as heart rate variability and temperature curve but also behavioral changes.

The authors acknowledge limitation of the study such as single bipolar EEG recording and postural effects on ICP. Nonetheless, this is an important disease model and techniques that can be used (perhaps with further updates to address the limitation) to better understand disease process and to develop treatments.

This reviewer has additional, mostly minor, suggestions and questions as follows:

1. Line 104-105: Rewording of the sentence is recommended to improve clarity. For example, it may be revised to "However, these studies have technical limitations that may affect inter-rater reliability and reproducibility."

2. In the introduction, it is not clear why continuous ICP monitoring would be important in alphavirus disease model. There are other methods that can be used to monitor a disease state "continuously and objectively" other than described brain monitoring devices, such as continuous EKG/telemetry and pulse oximeter for vitals. Thus, this reviewer recommends emphasizing the importance of continuous monitoring of brain-specific parameters based on a typical clinical course in humans in the introduction. For example, previous case reports of human EEEV included diffuse brain swelling as radiographic findings (N Engl J Med 1997; 336:1867-1874) and increased ICPs (Neurocrit Care. 2013 Aug;19(1):111-5, Emerg Infect Dis. 2016 Dec; 22(12): 2216–2217.). Particularly, rise in ICP and brain swelling can be a delayed finding that is observed a few days into the disease course, which argues for the need for continuous monitoring so that a timely treatment can be initiated. These are briefly mentioned in the later section under ICP monitor data, which may be introduced earlier to help readers follow the logic.

3. Similarly, the need for continuous EEG monitoring and in particular circadian index is unclear. Most clinicians use EEG to detect seizures even though it can also provide information about integrity of circadian rhythms. Is there evidence in alphavirus infection (or other similar diseases) that loss of circadian variation in EEG frequencies heralds or correlates with onset of important clinical changes (eg. seizure, brain swelling)? Why this particular focus is important needs further explanation.

4. It is described that infected macaques exhibit seizures. While EEG can be used for power frequency analysis and circadian index calculation, the most common clinical use of EEG is to detect seizures. Please describe how seizures were defined (e.g. behaviorally on human observation, per EEG findings, or combination), and how each seizure episode affected subsequent EEG findings (e.g. post-ictal slowing/increase in delta power), and whether any anti-seizure medication was attempted. Detailed description of seizures would also be important, regarding semiology (unilateral limb shaking, facial twitching, generalized shaking, staring, drooling, incontinence, etc.), duration, and frequency over the course of disease after viral infection. Seizures in human transiently increase ICP, and secondary cascade can lead to accelerated brain swelling and fatal rise in ICP (similar to what is described here). Therefore, the relative timing of observed seizure in ICP tracing is a clinically relevant information. As the authors mention in the limitation, a single lead EEG will have limited use in detecting seizures, especially focal seizures that have been described in human EEEV cases. However, if the field caused by the seizure is large enough, it likely affects the frequency bands of the single-lead EEG.

5. Line 446-448: This reviewer disagrees with the statement that monitoring of ICP serves an intuitive and proximal indicator of the response to infection within the brain, especially when an extradural ICP monitor is used. ICP reflects the pressure within the system (in human, ICP tracing commonly reflects pressure transmitted through the CSF space with the use of endoventricular drain), and increased ICP is more commonly a delayed and indirect finding of brain injury (which ultimately leads to inflammation and swelling) than a proximal measure of brain response. In addition, focal abnormalities in the deep brain structures such as thalami and lenticular nuclei have been reported in EEEV encephalitis, which potentially affect the venous drainage and CSF circulation, resulting in secondary rise in ICP that may be picked up by extradural ICP monitor once enough pressure builds up. A revision of the statement is recommended to avoid making controversial points.

6. Line 542-544: This reviewer disagrees with the statement that EEG is a direct measure of brain function. EEG is a electrophysiological marker of overall brain activity and does not measure any specific function of the brain. A revision is recommended, such as "The continuous EEG monitoring over multiple days in mock-infected animals allowed us to characterize normal variations across time of the day, reflecting normally present biological rhythms in different frequency ranges of brain activity."

Reviewer #2: The authors investigate a radiotelemetry to advance NHP animal model development. The manuscript details surgical techniques to implant devices and continuously and simultaneously monitor important physiological parameters (EEG and ICP). The data shows that both EEG and ICP are considerably altered post infection with EEEV or VEEV. The study is an important advance in the next generation of NHP model development.

Minors edits are suggested below.

1) Line 90-92. Rhesus Macaques were utilized in the study referenced. Also the first study of EEEV in NHP was the reference below.

E. Weston Hurst. Infection of therhesus monkey (Macaca mulatta) and the guinea-pig with the virus of equine encephalomyelitis. The Journal of Pathology and Bacteriology. 1936.

2) Line 531 the 1st EEEV studies were in Rhesus macaques. Please see the reference above. Also reference 48 is not correct, as there is no mention of NHP in the study.

3) Line 74 “Alphaviruses” is not a genus. The genus Alphavirus is in the Togaviridae family.

4) Line 75 “New World”. Remove quotation sign.

5) Line 83, remove “,” after natural.

6) Line 86, please reword sentence is awkward.

6. PLOS authors have the option to publish the peer review history of their article (what does this mean?). If published, this will include your full peer review and any attached files.

Reviewer #1: Yes: Minjee Kim

Reviewer #2: Yes: FAROOQ NASAR

---

## [Author Response · Author response to Decision Letter 0]

27 May 2020

We have attached a detailed response to the reviewers in the attachments.

---

## [Editor Report · Decision Letter 1]

1 Jun 2020

Applications of minimally invasive multimodal telemetry for continuous monitoring of brain function and intracranial pressure in macaques with acute viral encephalitis.

PONE-D-20-10011R1

Dear Dr. Teichert,

We are pleased to inform you that your manuscript has been judged scientifically suitable for publication and will be formally accepted for publication once it complies with all outstanding technical requirements.

With kind regards,

Negin P. Martin, Ph.D.

Academic Editor

PLOS ONE
---

## [Editor Report · Acceptance letter]

16 Jun 2020

PONE-D-20-10011R1 

Applications of minimally invasive multimodal telemetry for continuous monitoring of brain function and intracranial pressure in macaques with acute viral encephalitis. 

Dear Dr. Teichert:

I'm pleased to inform you that your manuscript has been deemed suitable for publication in PLOS ONE. Congratulations! Your manuscript is now with our production department. 

Kind regards, 

on behalf of

Dr. Negin P. Martin 

Academic Editor

PLOS ONE